# Overview of the Engagement Process to Develop the Future of Cancer Impact (FOCI) Report in Alberta: The Power of Collective Action

Anna Pujadas Botey [1,2,*], Tara R. Bond [3], Eliya Farah [4], Chantelle Carbonell [4], Stacey Dyck [5], Angela Estey [3], Douglas A. Stewart [1], Darren R. Brenner [4] and Paula J. Robson [2,3,6]

1    Cancer Strategic Clinical Network, Alberta Health Services, Calgary, AB T2N 2T9, Canada; douglas.stewart@ahs.ca
2    School of Public Health, University of Alberta, Edmonton, AB T6G 1C9, Canada; paula.robson@ahs.ca
3    Cancer Strategic Clinical Network, Alberta Health Services, Edmonton, AB T5J 3H1, Canada; tara.bond@ahs.ca (T.R.B.); angela.estey@ahs.ca (A.E.)
4    Department of Oncology, University of Calgary, Calgary, AB T2N 2T9, Canada; eliya.farah1@ucalgary.ca (E.F.); chantelle.carbonell@ucalgary.ca (C.C.); darren.brenner@ucalgary.ca (D.R.B.)
5    SJD Communications, Calgary, AB T2V 2T1, Canada; stacey@sjdcommunications.ca
6    Cancer Research & Analytics, Cancer Care Alberta, Alberta Health Services, Edmonton, AB T5J 3H1, Canada
*    Correspondence: anna.pujadasbotey@ahs.ca

**Abstract:** This commentary provides a detailed overview of the extensive stakeholder engagement efforts critical to the development of the Future of Cancer Impact (FOCI) in Alberta report. The overarching aim of the FOCI report was to support informed and strategic discussions and actions that will help key stakeholders in the province prepare for a future with increasing cancer incidence and survival. Employing a comprehensive approach and a diverse range of engagement activities, insights from a wide spectrum of stakeholders were gathered and subsequently used to shape the content of the report. This inclusive process ensured broad representation of perspectives, contributing to a deeper understanding of the complexities in cancer care. The outcome is a robust, consensus-driven report with recommendations set to drive significant transformations within the healthcare system. These efforts highlight the critical role of extensive, inclusive, and collaborative engagement in shaping healthcare initiatives and advancing discussions crucial for the future of cancer care in Alberta.

**Keywords:** future cancer impact; cancer care; stakeholder engagement; innovation; quality improvement; collaborative approach; collective action

## 1. Introduction

Initiated by Alberta Health Services' Cancer Strategic Clinical Network[TM] (SCN), the Future of Cancer Impact (FOCI) in Alberta report [1] offers a comprehensive exploration of cancer and cancer care in the province. It was designed to stimulate well-informed and strategic discussion and action around the future of cancer care in Alberta over the next two decades. Motivated by the evolution of cancer care since Alberta Health's Changing our Future: Alberta's Cancer Plan to 2030 [2] and in alignment with national priorities [3,4], the FOCI initiative builds upon the existing framework of cancer care in Alberta. Within this framework, cancer treatment is provided primarily through Cancer Care Alberta, supplemented by an intricate network of other healthcare providers both within and outside Alberta Health Services. Led by Dr. Paula Robson, Scientific Director of the Alberta Health Services' Cancer SCN, and Dr. Darren Brenner, Associate Professor at the University of Calgary, the FOCI report provides updated projections of key cancer statistics including incidence, mortality, and prevalence, and an overview of how cancer services are organized in Alberta today, alongside insights into models of cancer care and health equity. The report

includes recommendations for integrated actions that will help prepare Alberta to address future challenges in cancer care, particularly considering the steady increases in numbers of people being diagnosed and living with cancer over the next two decades. Data trends and projections from the FOCI report can be found in Brenner, Carbonell [5], with an overview of the findings and recommendations available here [withdrawn for review process].

The FOCI initiative embodies the Cancer SCN's strong commitment to leveraging the collective strengths of the cancer care community for co-designing and implementing innovative solutions [6,7]. This approach involved close collaboration across a diverse group of stakeholders engaged in every aspect of cancer care, spanning the entire continuum from prevention to end-of-life. The inclusive and collaborative nature of this initiative ensured the integration of a wide range of perspectives, leading to a comprehensive understanding of the complexities of cancer care in Alberta. The resulting report, informed by these insights, includes robust, consensus-driven recommendations and fosters shared ownership among all stakeholders involved in cancer care.

This manuscript aims to provide a detailed account of the continuous, proactive, and inclusive engagement with a wide range of stakeholders in shaping the content of the FOCI report, from its inception to the final stages. Serving as an exemplar, it highlights the importance of collaborative and multifaceted stakeholder involvement in advancing strategic discussions and setting the direction for the future of cancer care in Alberta. Moreover, by providing an overview of the engagement process in the FOCI in Alberta initiative, this manuscript also sheds light on the potential for broader applicability and impact beyond Alberta's borders. It has the potential to inform similar endeavors in other jurisdictions grappling with complex healthcare landscapes, thereby fostering greater inclusivity, transparency, and effectiveness in decision-making processes.

## 2. Future of Cancer Impact (FOCI) in Alberta Stakeholder Engagement

### 2.1. Engagement Principles

FOCI engagement efforts were guided by the following principles, aligned with the International Association for Public Participation (IAP2) Spectrum of Public Participation framework [8]:

Continuous participation: Extensively engaging from the project's inception and maintaining active involvement throughout, facilitating ongoing dialogue and adaptation.

Proactive engagement: Implementing diverse methods to ensure comprehensive participation and in-depth stakeholder input.

Responsible adaptation to feedback: Actively listening and implementing changes based on stakeholder feedback, underscoring a commitment to acknowledging and including their perspectives.

Building consensus: Aiming to harmonize various viewpoints to reach decisions that reflect collective interests and needs.

Fostering meaningful connections: Creating a collaborative and trustful network of stakeholders involved, valuing the views of all participants, and integrating their contributions as essential to the project's success.

### 2.2. Key Participants

The successful execution of the FOCI engagement approach relied on the involvement of the following key participant groups. Interplay between key participants is presented in Figure 1.

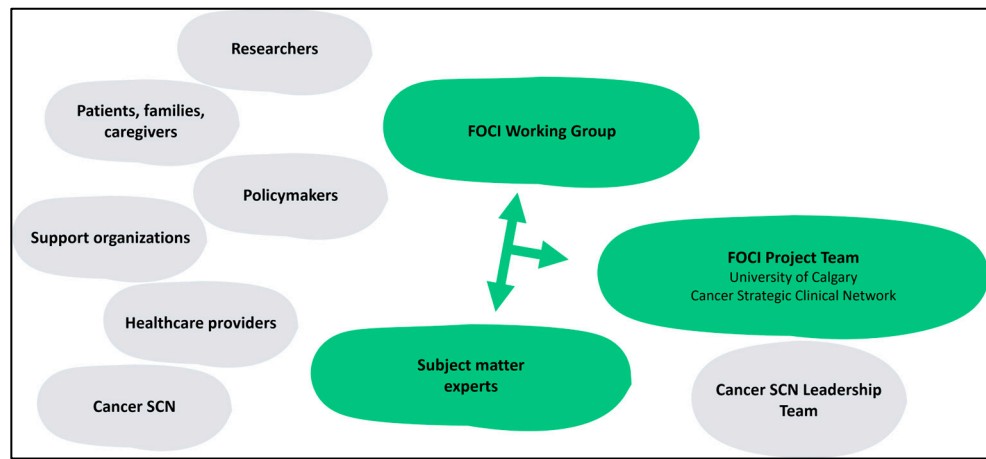

**Figure 1.** Interplay between key participants.

FOCI in Alberta Working Group: At the heart of the engagement efforts was the FOCI Working Group [9]. This group comprised individuals with diverse expertise in cancer care, including patients and their families and caregivers who have personal experiences with cancer and the healthcare system, as well as researchers, healthcare providers, and senior leaders and decision-makers related to the health system. The FOCI Working Group played a critical role in shaping the scope and parameters of FOCI, giving guidance from their diverse perspectives, and contributing valuable content and ideas. They reviewed drafts of the FOCI report for accuracy and validity and ensured that the perspectives and priorities they represented were integrated. The FOCI Working Group actively engaged in regular meetings, participated in ad hoc conversations, workshops, and working sessions, sharing insights and generating recommendations for action in various domains, reflecting a commitment to enhancing cancer care through a comprehensive and inclusive approach. A list of stakeholder groups represented in the FOCI Working Group is presented in Appendix A.

FOCI in Alberta Project Team: The FOCI Project Team, composed of members of the Cancer SCN and researchers affiliated with the University of Calgary, played a pivotal role throughout the engagement process. They were responsible for developing data projection models and detailed summaries that served as a foundational tool for the engagement. They also facilitated all engagement activities. This team actively solicited information, sought expert opinions, and conducted in-depth analyses to contribute to the collaborative and iterative content development process. The insights gained from these engagement activities informed the drafting of actionable recommendations, which were consolidated into the comprehensive FOCI report [1]. Additionally, the FOCI Project Team handled the dissemination of the report and advocated for its findings and recommendations. Regular touchpoints with the Cancer SCN Leadership Team ensured that the FOCI Project Team's work remained aligned with the broader goals of the Cancer SCN and facilitated connections with diverse stakeholders as required.

Subject matter experts: In addition to the aforementioned subject matter experts included in the FOCI Working Group, engagement expanded to include a broader range of specialists, across the complete spectrum of cancer care and population health. This diverse group included healthcare professionals, health system partners, researchers, policymakers, and representatives of support organizations. Patients, family members, and caregivers were also part of this group, offering unique insights into patient needs and experiences within the healthcare system. The involvement of subject matter experts was integral to the development of the FOCI report, as they significantly enriched its content. In addition to reviewing and contributing to specific report sections aligned with their areas of expertise, they actively participated in content validation processes, rigorously assessing the accuracy, comprehensiveness, and relevance of the information presented. Their dedication to this

validation process was invaluable, enhancing the overall credibility and utility of the FOCI report.

*2.3. Engagement Approach and Activities*

Collaborative content development approach: Recognizing the need to cast a wide net of stakeholders to gather diverse input and insights, the FOCI Project Team engaged with the members of the FOCI Working Group and specific subject matter experts related to the different areas included in the report. The engagement included a broad spectrum of participants, ranging from clinicians, healthcare providers, and researchers to patients, family members, caregivers, health system administrators, operational leads, non-profit and local organizations, and policymakers. The FOCI Project Team actively solicited information, sought expert opinions and data, and conducted in-depth analyses. Stakeholder engagement was facilitated through multiple means, encompassing small and large group meetings, workshops, content development sessions, a survey, a webinar, and ongoing communication through email and phone. Importantly, content contributors independently initiated and managed engagement activities within their respective stakeholder communities, including email conversations, focused group discussions, and informal gatherings.

Contributions stemming from these engagements underwent rigorous review and refinement processes during additional sessions involving the FOCI Working Group and the Cancer SCN Leadership Team. Representation of procedural steps involved in this collaborative content development process, operating within an iterative and feedback-driven model, is presented in Figure 2. The process resulted in a detailed and lengthy report, segmented into three sections encompassing 14 chapters and 46 recommendations [1]. Serving as a comprehensive repository, this document captures a wealth of information, data, and varied perspectives, addressing the changing needs and priorities for Alberta's cancer care system. To enhance accessibility, the extensive report was later distilled into more digestible formats, including a summary of key recommendations [10], infographics, and various presentations, thereby broadening its reach and utility.

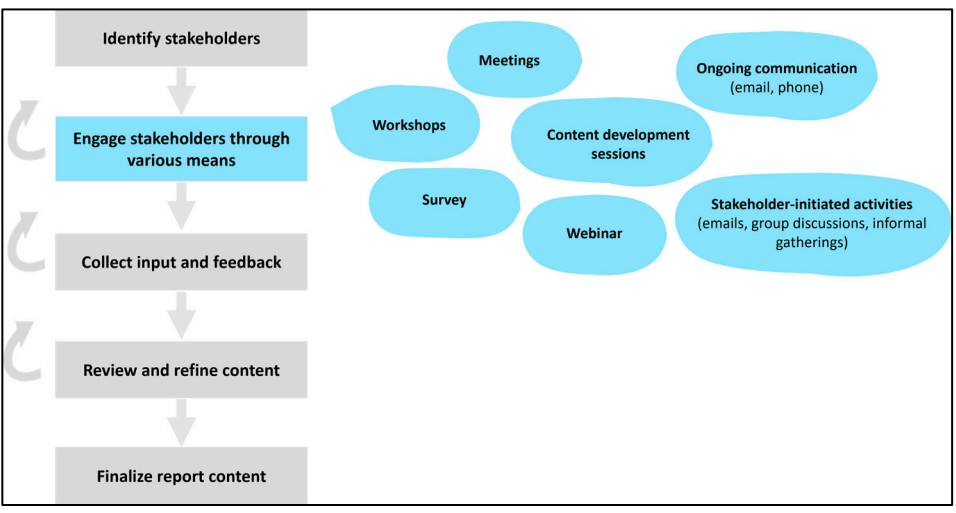

**Figure 2.** Procedural steps involved in the content development process.

Survey for feedback: An online survey was developed as a tool to gather additional feedback from interested individuals who had not been engaged in the development and writing of the FOCI report. Ensuring respondent confidentiality, the survey was designed to be anonymous, allowing participants to share their opinions freely. Developed using Qualtrics, the survey was crafted based on the key findings and recommendations outlined in the draft FOCI report. Respondents were provided with a URL to access a draft version of both the full FOCI report and a summary document [1,10], with encouragement to review them before completing the survey. The survey's design allowed respondents to provide

feedback on self-selected sections of the report. Within each section, the survey covered key findings, recommendations, and priorities for each chapter, enabling respondents to express their opinions on various aspects of the report's content. Additionally, respondents were prompted to indicate the timing they deemed most appropriate for each recommendation, with options ranging from Immediate (<1 year) to Mid-term (1–5 years), Long-term (>5 years), and Unsure. The survey was distributed in November and December 2022 to over 1300 individuals across the province, including clinicians, allied health professionals in cancer care, the provincial research community, patients, caregivers, policymakers, and representatives of relevant government and non-governmental agencies. To further expand the engagement process, participants were encouraged to share the survey invitations with others or recommend additional participants, who were also invited to participate. This approach effectively broadened our reach, resulting in a total of 202 responses.

Webinar for outreach: As part of the outreach endeavors, the FOCI Project Team organized a webinar led by Dr. Darren Brenner, the project lead for the FOCI initiative and co-chair of the FOCI Working Group [11]. This webinar played a dual role; firstly, it disseminated key findings from the draft report, providing participants with an in-depth understanding of the current state of cancer care, and secondly, it served as an important platform to promote the survey as a crucial feedback mechanism. During the presentation, Dr. Brenner not only highlighted significant aspects of the report but also shared preliminary survey results, offering attendees a real-time glimpse into the ongoing engagement process. This strategic sharing of survey data underscored the collective impact of stakeholder feedback on shaping the report. The session fostered an interactive environment, encouraging attendees to ask questions and share perspectives, thereby enriching the dialogue and deepening the understanding of issues. This approach not only stimulated thoughtful discussions but also ensured that a comprehensive and diverse range of viewpoints were considered in refining the final report. To enhance accessibility and inclusivity, the webinar was recorded and made available online, extending the reach of the FOCI initiative's outreach efforts and enabling broader participation in this critical conversation.

Strategic advocacy meetings: Strategic advocacy meetings led by the Cancer SCN Leadership Team marked the culmination of the engagement process facilitated by the FOCI Project Team to develop the content of the FOCI report. Their primary goal was to support the insights gathered from extensive stakeholder engagement and emphasize the importance of the report's recommendations. These recommendations, developed through an inclusive process involving diverse stakeholder input and validation, now form the basis for discussions and actions involving specific entities such as philanthropic organizations, Cancer Care Alberta, and other influential cancer care stakeholders. This strategic dissemination approach aims to ensure that the report's insights and recommendations create a significant impact on the direction of cancer care in Alberta. The active involvement and endorsement of these stakeholders highlights the collaborative and inclusive nature of this initiative, ultimately driving transformative changes and innovations within the healthcare system.

## 3. Conclusions

Effective stakeholder engagement has been the cornerstone of the FOCI initiative, playing a crucial role in guiding the development of the FOCI report, shaping its insights, and setting the stage for transformative change in cancer care in Alberta. Central to this success was the dedicated FOCI Project Team, whose focused and sustained efforts were instrumental in facilitating all engagement activities. Their commitment to employing a diverse array of engagement strategies ensured broad and inclusive participation. A non-linear and unrestricted approach to engagement empowered stakeholders to take initiative, tapping into their networks and leading their own feedback processes, thereby further enriching the insights gathered.

The collaborative efforts of the FOCI Project Team, along with the inclusion of the FOCI Working Group and the invaluable contributions of subject matter experts and stakeholders

from diverse backgrounds, have collectively resulted in a nuanced discussion of Alberta's current and future cancer care landscape. The engagement process has transformed the FOCI report from a document of limited impact into a catalyst for ongoing dialogue and strategy development to address the future challenges and opportunities in cancer care in Alberta. While the focus of the FOCI initiative has been on addressing the specific challenges within Alberta's cancer care landscape, we recognize that the approach, the breadth and type of stakeholders and experts involved, and the insights developed through this endeavor hold the potential for broader applicability and impact beyond the borders of Alberta. This comprehensive work can serve as a valuable resource and model for healthcare professionals, policymakers, and stakeholders globally who are striving to enhance cancer care delivery and outcomes in their respective regions.

We extend our gratitude to all stakeholders for their enthusiastic support and contributions. Looking forward, we are committed to continuing this partnership, leveraging diverse perspectives to navigate the challenges ahead in cancer care. The FOCI initiative is dedicated to ensuring that Albertans facing cancer receive the best possible care and support, moving us closer to realizing the vision outlined in the report.

**Author Contributions:** Conceptualization, A.P.B., T.R.B., E.F., C.C., A.E., D.A.S., D.R.B. and P.J.R.; project administration, A.P.B., T.R.B., E.F., C.C. and D.R.B.; supervision, D.R.B. and P.J.R.; visualization, A.P.B.; writing—original draft preparation, A.P.B. and S.D.; writing—review and editing, A.P.B., T.R.B., E.F., C.C., S.D., A.E. and D.A.S.; the FOCI Working Group, D.R.B. and P.J.R. All authors have read and agreed to the published version of the manuscript.

**Funding:** This research received no external funding.

**Data Availability Statement:** The original contributions presented in the study are included in the article; further inquiries can be directed to the corresponding author.

**Acknowledgments:** We would like to thank the FOCI working group members and everybody who participated in the FOCI engagement efforts for their input to the development of the FOCI report.

**Conflicts of Interest:** The authors declare no conflicts of interest. Stacey Dyck acknowledges her affiliation with SJD Communications. However, this affiliation did not influence the design, execution, or reporting of the work presented in this manuscript. Dyck's involvement with SJD Communications is unrelated to the subject matter discussed herein, and there is no financial or personal gain associated with the findings presented. The authors affirm that this manuscript was prepared in an objective and unbiased manner.

## Appendix A. Membership of FOCI in Alberta Working Group

The Working Group was co-chaired by a member of the Cancer Strategic Clinical Network and a member of the University of Calgary, and comprised stakeholders with broad representation from the following groups:

- Cancer Strategic Clinical Network, Alberta Health Services;
- Researchers, University of Calgary and University of Alberta;
- Patient, family, and caregiver partners;
- Alberta's Tomorrow Project, Alberta Health Services;
- Analytics and Performance Reporting and Health Economics, Alberta Health;
- Canadian Partnership Against Cancer;
- Cancer Prevention and Screening Innovation (formerly Alberta Cancer Prevention Legacy Fund), Alberta Health Services;
- Community Oncology, Alberta Health Services;
- Diagnostic Imaging, Alberta Health Services;
- Health Economics, Alberta Health Services;
- Health Innovation, Alberta Health;
- Lab Services, Alberta Precision Laboratories;
- Pharmacy, Alberta Health Services;
- Primary Health Care Integration Network, Alberta Health Services;

- Provincial Population and Public Health (formerly Population and Public Health Strategic Clinical Network), Alberta Health Services.

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
