# Peer review of "Overview of the Engagement Process to Develop the Future of Cancer Impact (FOCI) Report in Alberta: The Power of Collective Action"

_curroncol, doi:10.3390/curroncol31030111_

Round 1

Reviewer 1 Report

Comments and Suggestions for Authors

Thank you for the opportunity to review this informative overview of a strategic planning process and document of cancer care in Alberta. I appreciate this overview, but I have a few suggestions.

1. For Section 2.1., how did you arrive at these engagement principles? Are they supported by any specific theories or frameworks?

2. I am curious about the grey rim of the FOCI Working Group, Subject matter experts, and FOCI Project Team in Figure 1. Are they supposed to symbolize something?

3. I think Section 2.2. can be reduced by inserting another figure. 

4. In addition to describing all the activities for engagement (Section 2.3), I think a figure that graphically illustrates relationships or procedural steps of these activities will be helpful.

Reviewer 2 Report

Comments and Suggestions for Authors

The authors have set out to detail the process of what was done and whom was involved in the FOCI initiative/report. It sounds like the final outcomes have been submitted for review outside of this commentary. It is well written and flows well.

My only concern is how applicable it would be to the wider audience. It is very one province-centric, and without any data to review, there is no discussion on how the process in Alberta could be applied to other jurisdictions. I think some work needs to be done on the conclusions to make this more enticing for the readers at large who are not in Alberta. 

Round 2

Reviewer 1 Report

Comments and Suggestions for Authors

Thank you for your consideration of my suggested edits.

Author Response

Thank you for your valuable insights and suggestions.

Reviewer 2 Report

Comments and Suggestions for Authors

The authors have answered my questions. No issues.

Author Response

(The authors gave the same response as above.)
